# Conjugates of Methylene Blue with Cycloalkaneindoles as New Multifunctional Agents for Potential Treatment of Neurodegenerative Disease

**DOI:** 10.3390/ijms232213925

**Published:** 2022-11-11

**Authors:** Sergey O. Bachurin, Elena F. Shevtsova, Galina F. Makhaeva, Alexey Yu. Aksinenko, Vladimir V. Grigoriev, Tatiana V. Goreva, Tatiana A. Epishina, Nadezhda V. Kovaleva, Natalia P. Boltneva, Sofya V. Lushchekina, Elena V. Rudakova, Darya V. Vinogradova, Pavel N. Shevtsov, Elena A. Pushkareva, Ludmila G. Dubova, Tatiana P. Serkova, Ivan M. Veselov, Vladimir P. Fisenko, Rudy J. Richardson

**Affiliations:** 1Institute of Physiologically Active Compounds at Federal Research Center of Problems of Chemical Physics and Medicinal Chemistry, Russian Academy of Sciences, 142432 Chernogolovka, Russia; 2Emanuel Institute of Biochemical Physics, Russian Academy of Sciences, 119334 Moscow, Russia; 3Department of Pharmacology, Institute of Biodesign and Complex System Modelling, Biomedical Science & Technology Park, Sechenov I.M. First Moscow State Medical University, 119991 Moscow, Russia; 4Department of Neurology, University of Michigan, Ann Arbor, MI 48109, USA

**Keywords:** conjugates, methylene blue, cycloalkaneindoles, multifunctional agents, neurodegenerative disease, cholinesterases, mitochondria, tubulin, NMDA-receptor, neuroprotection

## Abstract

The development of multi-target-directed ligands (MTDLs) would provide effective therapy of neurodegenerative diseases (ND) with complex and nonclear pathogenesis. A promising method to create such potential drugs is combining neuroactive pharmacophoric groups acting on different biotargets involved in the pathogenesis of ND. We developed a synthetic algorithm for the conjugation of indole derivatives and methylene blue (**MB**), which are pharmacophoric ligands that act on the key stages of pathogenesis. We synthesized hybrid structures and performed a comprehensive screening for a specific set of biotargets participating in the pathogenesis of ND (i.e., cholinesterases, NMDA receptor, mitochondria, and microtubules assembly). The results of the screening study enabled us to find two lead compounds (**4h** and **4i)** which effectively inhibited cholinesterases and bound to the AChE PAS, possessed antioxidant activity, and stimulated the assembly of microtubules. One of them (**4i**) exhibited activity as a ligand for the ifenprodil-specific site of the NMDA receptor. In addition, this lead compound was able to bypass the inhibition of complex I and prevent calcium-induced mitochondrial depolarization, suggesting a neuroprotective property that was confirmed using a cellular calcium overload model of neurodegeneration. Thus, these new **MB**-cycloalkaneindole conjugates constitute a promising class of compounds for the development of multitarget neuroprotective drugs which simultaneously act on several targets, thereby providing cognitive stimulating, neuroprotective, and disease-modifying effects.

## 1. Introduction

The growing number of patients with neurodegenerative diseases, such as Alzheimer’s disease (AD) and Parkinson’s disease (PD) (which are already highly prevalent and increasing) and amyotrophic lateral sclerosis (ALS) (which is low but increasingly prevalent), especially in economically developed countries, makes the search for drugs against these diseases a highly relevant task. The few drugs that are used to treat AD are predominantly symptomatic, i.e., they partially compensate for the decrease in cognitive functions and memory by forced activation of neurotransmitter systems, but they do not influence the neurodegeneration process itself. Currently, the key approach to the development of new-generation drugs for the treatment of AD and ALS is the search and design of substances capable of retarding the progression of neurodegeneration (i.e., disease-modifying agents). It is expected that the development of drugs that act simultaneously on several biological structures participating in neuron death and in the loss of neuron functional activity, or so-called ***multi-target**-directed ligands (MTDLs)***, would provide more effective therapy of neurodegenerative diseases directed toward the pathogenesis of the disease [1,2,3]. A promising method to address this problem is the design of new pharmaceutical agents that contain distinct neuroactive pharmacophoric groups acting on different biotargets involved in the pathogenesis of neurodegenerative diseases.

Previously, we developed a synthetic algorithm for the conjugation of several pharmacophoric ligands that act on the key stages of pathogenesis, synthesized hybrid structures, and performed screening for a broad range of biotargets participating in the pathogenesis of AD [4,5,6].

The present communication describes the development of another series of conjugates and an evaluation of their possible therapeutic capability. These conjugates comprise indole derivatives and methylene blue, which is a pharmacophore with a unique neuroprotective potential (see Figure 1).

Methylene blue (**MB**) belongs to the family of phenothiazines, which have a broad range of therapeutic and diagnostic applications. A significant number of MB and derivatives with a wide spectrum of biological activity, including the effect on targets important for the treatment of Alzheimer’s disease, were created [7,8].

Owing to both cationic and lipophilic properties, **MB** easily penetrates the blood–brain barrier, binds to the mitochondrial membrane, and diffuses to the mitochondrial matrix, where it forms, in low doses, a redox equilibrium with electron transport chain enzymes, thus enhancing mitochondrial respiration. Thus, **MB** is a potent redox agent that is highly bioavailable for mitochondria, which reduces the frequency of the formation of mitochondrial reactive oxygen species (ROS); hence, it can decrease the rate of cell aging [9,10]. Furthermore, **MB** modifies the activity of some membrane-bound transporters and ion channels—in particular, the voltage-dependent Na^+^ channel and the Ca^2+^-dependent K^+^ channel—and this is considered to be responsible for the depolarizing effect of MB on the neuronal membrane [11,12]. Correspondingly, MB affects neuronal relationships by changing the activity of cholinergic, monoaminergic, or glutamatergic synaptic neurotransmission [13,14,15]. In addition, it has been shown that **MB** enhances memory in a normal brain, which can be caused by its effect on neurotransmission. However, many researchers also attribute this effect to the stimulating action of **MB** on mitochondria [16]. According to recent studies, **MB** has a high potential for the treatment of ischemic and hypoxic disorders, AD, or some other neurodegenerative diseases, owing to both its neuroprotective potential and its cognitive stimulation effect [17,18,19,20]. It has also been shown that **MB** considerably decreases the level and the toxicity of abnormal protein aggregates (Huntingtin protein, α-synuclein, tau protein, and β-amyloid). However, the mechanisms of this activity are varied, including redox inhibition of aggregation, decreasing oligomer levels by promoting fibril formation, photodynamic destruction of aggregates, and increasing proteasome activity [21,22,23,24,25]. One more argument for our choice of this pharmacophore is derived from the earlier observation that phenothiazine derivatives, including **MB**, can efficiently inhibit cholinesterase enzymes [26,27,28,29,30,31].

Meanwhile, many indoles and carbazole derivatives have neuroactive properties. For example, aminopropylcarbazole derivatives exhibit pro-neurogenic and neuroprotective activities without obvious toxicity, with the inhibition of the mitochondrial permeability transition being one of the key mechanisms of such activity [32]. In 2013, Zhu et al. synthesized a series of N-substituted carbazoles. Lead compounds were identified among these products. When present in a concentration of only 3 µM, these compounds had a neuroprotective action on HT-22 neuronal cells against glutamate- or homocysteic acid-induced cell injury. The neuroprotective effect was attributed to the antioxidant activity mediated by a GSH-independent mechanism [33]. Some cyanine compounds based on carbazole were found to prevent Aβ aggregation, down-regulate the activity of glycogen kinase synthase-3β (GSK-3β), and decrease the hyperphosphorylation of tau protein in the triple transgenic mouse model of AD [34]. Therefore, we chose cycloalkaneindoles, a representative of which is tetrahydrocarbazole, as the second pharmacophore for the design of multifunctional conjugates.

## 2. Results and Discussion

### 2.1. Chemistry

The target conjugates of **MB** and cycloalkaneindoles linked with a 1-oxopropylene spacer were synthesized, as shown in Figure 1, by the reaction of 1-[3,7-bis(dimethylamino)phenothiazin-10-yl]propenone **1** with cycloalkaneindoles **2**. Detailed synthesis and NMR spectroscopy data for conjugates **3a–n** and **4a–n** were reported previously [35].

The prepared conjugates were subjected to primary screening using the methodology relevant to the treatment of neurodegenerative diseases that we proposed previously [6]. The system included neurotransmitter targets related to the compensation of cognitive functions, including: cholinesterases and glutamate receptors; mitochondria and the prevention of the mitochondrial permeability transition, which may provide cyto- and neuroprotection; and microtubules, whose destabilization is a specific feature of certain neurodegenerative diseases, e.g., tauopathies.

### 2.2. Study of the Esterase Profile of Conjugates **4** and Their Ability to Displace Propidium from the Peripheral Anionic Site of Acetylcholinesterase

We used enzyme kinetics and molecular docking to study the inhibitory activity of the **MB**-cycloalkaneindole conjugates against acetylcholinesterase (EC 3.1.1.7, AChE), butyrylcholinesterase (EC 3.1.1.8, BChE), and the structurally related carboxylesterase enzyme carboxylesterase (EC 3.1.1.1, CES). In addition, the ability of these compounds to competitively displace propidium iodide (PI), a selective ligand of the AChE peripheral anionic site (PAS) which binds β-amyloid, was studied by a fluorescent method to indirectly evaluate the test compounds as inhibitors of the pro- β-amyloid aggregation activity of AChE.

AChE and BChE can hydrolyze the neurotransmitter acetylcholine, and they are important targets for the amelioration of cognitive symptoms during the development of AD and/or AD-like dementia [36]. BChE inhibition is especially useful against stages of the disease when BChE has taken on part of the role of acetylcholine hydrolysis that has been compromised by a decrease in AChE activity [37,38,39]. For this reason, it is thought that compounds that inhibit both AChE and BChE increase the efficiency of treatment [40,41].

CES is responsible for the hydrolysis of numerous drugs containing ester groups [42], and the inhibition of this enzyme by anticholinesterase drugs taken by AD patients may lead to adverse drug-drug interactions [43,44]. Our approach included the determination of the esterase profiles of compounds, i.e., the comparative determination of their inhibitory activities against a few structurally related serine hydrolases [30,43,44,45,46,47,48,49], making it possible to evaluate the ability of the compounds to inhibit cholinesterases and to reveal their potential adverse effects in an early stage of research.

#### 2.2.1. Inhibitory Activity of Conjugates 4 against Human Erythrocyte AChE, Equine Serum BChE, and Porcine Liver CES

The esterase profile was estimated using the human erythrocyteAChE, the equine serum BChE, and the porcine liver CES. The equine and porcine enzymes have a high degree of sequence identify with their human counterparts, and the applicability of these enzymes for determining the esterase profile of new compounds has been shown in our previous studies [26,35,49,50]. The inhibitory activity was characterized as the percentage of inhibition at 20 µM concentration or by determining the IC_50_ value, i.e., the inhibitor concentration needed to decrease the enzyme activity by 50%. As reference compounds, we used the base pharmacophores: carbazole, **MB**, and reduced **MB** (leuco-**MB**, **MBH_2_**).

The results, summarized in Table 1, show that all conjugates **4** (Figure 1) inhibit CES to a low extent, but they have relatively high inhibitory activities against AChE and BChE. The inhibition of AChE and BChE occurrs in the micromolar range, with pronounces selectivity for AChE. Note that conjugates **4** inhibit AChE and BChE at the level of the parent pharmacophores or somewhat below this level. The inhibitory activity depends little on the ring size.

#### 2.2.2. Study of the Mechanism of AChE Inhibition by Conjugates **4**

The mechanism of AChE inhibition by MB-cycloalkaneindole conjugates was considered in relation to three of the most active compounds: **4c, 4i**, and **4m**. The results were analyzed using the Lineweaver–Burk double reciprocal plot, and the graphical analysis of the kinetic data is shown in Figure 1. The binding of **4c (a), 4i (b)**, or **4m (c)** to AChE changed both the *V*_max_ and the *K*_m_, which is typical of mixed type inhibition. The AChE inhibition constants by compounds **4c** (*K_i_* = 1.98 ± 0.17 µM for the competitive component and *αK_i_* = 5.83 ± 0.31 µM for the noncompetitive component), **4i** (*K_i_* = 1.12 ± 0.05 µM, *αK_i_* = 3.19 ± 0.10 µM), and **4m** (*K_i_* = 1.06 ± 0.04 µM, *αK_i_* = 4.03 ± 0.15 µM) were determined. These conjugates were found to be effective reversible mixed-type inhibitors of AChE.

#### 2.2.3. Molecular Modeling

The kinetic data were in full agreement with the molecular docking results. Molecular docking into AChE was performed for the same compounds from the kinetics studies, including **4c** and **4m**, which differed according to the size of the methylene ring (*n* = 2 and *n* = 4, respectively), and compound **4i**, which contained a CF_3_O substituent and was the most potent AChE inhibitor.

As was shown in our earlier studies, the choice of the X-ray diffraction structure for molecular docking can significantly affect the modeling results [51]. Based on these previous findings, we used two X-ray structures of hAChE as docking targets in the present study [52]. For the *apo*-hAChE structure (PDB ID 4EY4), the ligands were found only in the PAS. For the X-ray structure of hAChE co-crystallized with donepezil (PDB ID 4EY7), the ligands were found predominantly in the catalytic active site (CAS). These two main protein structures had different conformations of the Tyr337 side chain which intercepted the enzyme gorge with the *apo*-state and acted as one of the valves of the bottleneck. In the X-ray structure with donepezil in the gorge, Tyr337 was rotated and moved to the wall, making room for a ligand, which made this structure (PDB ID 4EY7) a preferable target for the molecular docking of bulky ligands.

The positions of compounds in the CAS had a specific π-π interaction between the MB moiety and the Trp86 ring (Figure 2a,b). In both positions of compound **4c**, with the methylene blue moiety in the PAS and the CAS, the cycloalkaneindole ring was squeezed between Tyr341 and Trp286 in the PAS, just above the bottleneck. For compound **4m**, a deeper position of the cycloalkaneindole ring was found. Thus, the docking results show that the ligands are dual-site inhibitors, which initially bind to the PAS and then move down the gorge after the opening of the bottleneck (according to the mechanism described earlier [53]) to the active site, where their binding is stabilized by π-π interaction between the Trp86 and the MB ring. There are X-ray structures of MB in complex with Torpedo californica AChE (TcAChE) obtained in the presence and absence of polyethylene glycol (PEG), respectively (PDB ID 5E4T and 5DLP, [54]). In these structures, MB is located in the PAS. However, it cannot serve as a reference for MB derivatives due to the significant change in its geometry upon conjugation to the cycloalkaneindole component. MB is completely planar, and joining the linker to the nitrogen atom impairs aromatic conjugation and leads to a loss of planarity in the MB group (Figure 2). This could be seen in the available X-ray structures of the MB conjugates (e.g., PDB ID 1LVJ, 4MA7, 5NUN). QM optimization of the compounds **4** revealed that the angle between the two halves of the MB fragment is 131°, which corresponds with the X-ray data.

The position of compound **4i** is noticeably different from the positions of compounds **4c** and **4m** due to the CF_3_O group (Figure 2c). It is surrounded by aromatic residues Tyr337 and Trp86. Because the carbon atom of this moiety is surrounded by electron-withdrawing atoms, its interactions with aromatic residues could be considered as a tetrel bond [55]. Similar configurations were observed for -CF_3_…π interactions in computational studies [56] and in experimental X-ray structures of protein–ligand complexes [57]. As a result of such interactions of the CF_3_O group, the **MB** moiety better occupies the PAS compared to the other compounds. Notably, the sulfur atom of the **MB** group is directed toward the Trp286 aromatic ring, which, along with the nearby Tyr72 residue, forms an entity called a “3-bridge cluster” [58,59].

#### 2.2.4. Inhibition of EeAChE and Displacement of Propidium Iodide from the EeAChE Peripheral Anionic Site

It is known that AChE plays an important role in β-amyloid processing via the PAS, which interacts with soluble peptides of β-amyloid and promotes their aggregation [60,61,62,63]. Therefore, the development of drugs that block the PAS of AChE, disrupt its interaction with β-amyloid, and decrease the AChE-induced β-amyloid aggregation is a promising trend of the anti-amyloid strategy of AD therapy.

We used a fluorescence method to determine the ability of the compounds to competitively displace propidium iodide (PI) from the AChE PAS. This procedure is widely known and often used as a primary screen to indirectly identify compounds as inhibitors of β-amyloid aggregation mediated the AChE PAS. The method is based on the increase in the fluorescence intensity of PI upon binding to AChE. The decrease in the fluorescence intensity of AChE-bound propidium in the presence of test compounds shows the ability of these compounds to displace propidium and to bind to the PAS of AChE. Donepezil and decamethonium were used as reference compounds.

We selected AChE from electric eels (*Electrophorus electricus*) for consistency with our other reports and because of its purity, specific activity, and lower cost compared to human AChE [64]. First, the inhibitory activity of conjugates **4** against *Ee*AChE, i.e., against its ability to hydrolyze acetylthiocholine, was investigated. The results are summarized in Table 2 and demonstrate high inhibitory activity of compounds against the catalytic activity of *Ee*AChE.

Following our confirmation of the inhibitory activity against *Ee*AChE, we then investigated all conjugates **4** for their ability to bind to the AChE PAS and competitively displace the selective ligand (PI). The results are summarized in Table 2. Conjugates **4** in a concentration of 3 µM decreased the fluorescence intensity by 10–20%, while at 20 µM, this decrease was 15–37%. Thus, these compounds displaced PI from the AChE PAS more efficiently than the reference compounds donepezil and decamethonium, and in some cases, they approached the activities of **MB** and **MBH_2_**.

The results indicate that the test compounds can effectively bind to the AChE PAS which mediates for β-amyloid aggregation. Moreover, these findings agree with the kinetics data (Figure 2) demonstrating mixed-type inhibition, as well as the molecular docking results, showing binding to the AChE PAS (Figure 2). These findings suggest that the studied conjugates have the potential to block the AChE-induced aggregation of β-amyloid.

### 2.3. Studies of the Antioxidant Activity of **MB**-Cycloalkaneindole Conjugates

#### 2.3.1. Studies of the Primary Antioxidant Activity of Conjugates **4**

The primary antioxidant activity of conjugates **4** was determined by their ability to scavenge free radicals in the ORAC-FL and ABTS assays. The ORAC-FL assay is based exclusively on the hydrogen atom transfer (HAT) mechanism, whereas the ABTS assay is a combined method in which both HAT and single electron transfer (SET) mechanisms are possible. Trolox was used as a reference antioxidant (the antioxidant activity of the test compounds was referred to the activity of Trolox). The well-known antioxidant catechol was used as a positive control.

##### Evaluation of the Antiradical Activity by the ABTS Radical Cation Scavenging Assay

The ABTS assay is based on the direct binding of a model ABTS radical cation (2,2′-azino-bis-(3-ethylbenzothiazoline-6-sulfonate), ABTS^•+^) by antioxidants [65]. The measurements were performed as previously described in detail [64,66].

The radical scavenging activity was expressed as TEAC values (Trolox equivalent antioxidant capacity) and determined as the ratio between the slopes obtained from the linear correlation of the ABTS radical absorbance with the concentrations of the test compounds and Trolox. For the test compounds, we also determined the IC_50_ values (the compound concentration [μM] required for a 50% decrease in the concentration of the ABTS radical). The results are summarized in Table 3.

The results indicate that all **MB**-cycloalkaneindole conjugates have a high ABTS^•+^- scavenging activity, which is close to or somewhat higher than the activity of the standard antioxidant Trolox (Table 3). The highest activity in this assay was found for compound **4e** (R=CH_3_O, R_1_=CH_3_): TEAC = 1.46.

##### Evaluation of the Oxygen Radical Absorbance Capacity by a Fluorescence Method (ORAC-FL)

The ability of conjugates **4** to decrease the amount of peroxyl radicals as an additional characteristic of primary antioxidant activity was determined by a fluorescence assay (ORAC-FL) using fluorescein (FL) as the fluorescence probe. The method is based on the measurement of the fluorescence intensity characterizing the degree of destruction of the fluorescence probe under the action of peroxyl radicals. In the presence of antioxidants, the degree of destruction of the fluorescence probe induced by peroxyl radicals decreased and, hence, the time of fluorescence increased.

The peroxyl radical scavenging capacity of the test compounds was characterized in terms of the Trolox equivalent (TE), which is equal to the ratio of the Trolox concentration to the concentration of a test compound when they show equal fluorescence intensities in the assay. The peroxyl radical scavenging capacity of Trolox was set as 1 [67,68]. The results of the ORAC-FL assay are summarized in Table 3.

It was shown that the **MB**-cycloalkaneindole conjugates possessed high peroxyl radical scavenging capacity, which markedly exceeded that of Trolox, being in the range from 4 to 15 TE. All conjugates containing 7- or 8-membered cycloalkane moieties were generally more efficient. Compound **4m** (*n* = 4, R=R_1_=H) was the most active in the ORAC test. Among the conjugates with 6-membered aliphatic rings (tetrahydrocarbazole), the highest activity was characteristic of compound **4e** (R=CH_3_O, R_1_=CH_3_), which was also the most efficient in the ABTS test.

The data on the radical scavenging capacity for **MB** and **MBH_2_** are not shown in Table 3 as the values for MB and MBH_2_ were not detectable with the ABTS and ORAC methods. This may be attributable to the very low redox potential of **MB** (11 mV) [69] and the ease of cycling between the oxidized and reduced forms.

##### Frontier Orbital Calculations

The DFT(B3LYP)/6-31++G ** quantum mechanical calculations carried out for **MB**- cycloalkaneindole conjugates resulted in the visualization of the frontier orbitals (Figure 3) and confirmed the experimental results of the antioxidant activity of the test compounds. The HOMO location on the cycloalkaneindole moiety (Figure 3) explains why the size of the aliphatic ring influences the antioxidant capacity in the ORAC assay. In particular, higher antiradical activity was demonstrated by compounds with 7- and 8-membered aliphatic rings in the cycloalkaneindole moiety.

#### 2.3.2. Inhibition of the Iron-Induced LP in Rat Brain Homogenate

The influence of **MB** and its conjugates with indole derivatives on Fe^3+^-induced lipid peroxidation (LP) in rat brain homogenate was studied over a broad concentration range (from 10 nM to 30 µM).

All the conjugates, as well as **MB** and **MBH_2_**, exhibited no pro-oxidant activity in the concentrations used and under our experimental conditions. However, as shown in Table 4, all compounds showed efficient concentration-dependent inhibition of Fe^3+^-induced LP in rat brain homogenate. The IC_50_ values for all the test conjugates were below 1.9 μM or IC_50_ values for MB and close to IC_50_ values for **MBH_2_** (Table 4). Effective inhibition of LP in the rat brain homogenate may be particularly connected to the high radical-scavenging activity of the compounds found in the ABTS and ORAC tests (Table 3). However, the lack of direct correlation between the radical scavenging activity in the model ABTS and ORAC assays and the biologically relevant inhibition of LP in rat brain homogenates may be indicative of additional contributions from other mechanisms.

### 2.4. Effect of **MB**-Cycloalkaneindoles Conjugates on Mitochondrial Functions

#### 2.4.1. Evaluation of the Effect of Test Compounds on the Membrane Potential and Calcium-Induced Mitochondrial Permeability Transition

The primary screening of new compounds to determine their effect on the mitochondrial membrane potential formed due to respiratory chain (RC) activity after additions of RC substrates was performed on isolated rat liver mitochondria. It can be used to estimate the probability of the potential toxicity of these compounds. Furthermore, the influence of the compounds on the calcium-dependent depolarization of isolated rat liver mitochondria in this assay made it possible to evaluate the effect on the mitochondrial permeability transition.

As shown in Table 4, in the presence of NADH-dependent substrates of the mitochondrial respiratory chain (RC) complex I, none of the newly synthesized conjugates in 1 μM concentration affected the mitochondrial membrane potential or calcium-induced depolarization. However, in the presence of the complex I inhibitor rotenone and a substrate of RC complex II, compound **4m**, as well as **MB** induced mitochondrial depolarization, which increased with time and enhanced calcium-dependent mitochondrial depolarization. When present in a concentration of 30 μM, all compounds depolarized mitochondria both in the presence of the substrates of complex I and, to a larger extent, in the presence of complex II substrate succinate and complex I inhibitor rotenone. The lowest degree of depolarization in the presence of the substrates of complex II and I was found in compounds **4i** and **4h**. In addition, in a 30 μM concentration, **MB** and **4i** completely suppress, while compound **4h** considerably decrease the calcium-induced depolarization, but not influence the depolarization induced by carbonyl cyanide-3-chlorophenylhydrazone (CCCP), which is an uncoupler of the mitochondrial respiratory chain.

Subsequently, a more detailed study of compound **4i** was carried out against a number of targets relevant for the selection of potential drugs for the treatment of neurodegenerative diseases. The results are presented in the following sections.

#### 2.4.2. Evaluation of the Effect of the Lead Compound **4i** on the Bioenergetic Potential of Rat Liver Mitochondria

**MB** possesses amphiphilic characteristics as well as unique electron-donating and electron-withdrawing properties. These features account for its ability to transfer electrons in the mitochondrial electron transport chain and its capacity to stimulate mitochondrial respiration, decrease the inhibition of respiratory chain complexes [70], and also provide an alternative electron transfer in the case of a dysfunctional electron transport chain [71].

The ability to bypass complex I inhibition was determined for **MB**, its reduced form MBH_2_, and compound **4i** (Figure 4).

The addition of rotenone to mitochondria energized by complex I substrates (glutamate/malate) causes considerable depolarization. The subsequent addition of MB or MBH_2_ induces concentration-dependent repolarization. Compound **4i** exhibits this type of activity only in a 10 µM concentration, and the effect is much less pronounced.

Meanwhile, compound **4i** can stimulate respiration of isolated rat liver mitochondria, which was estimated using an XFe96 Seahorse extracellular flux analyzer (Figure 5).

Analysis was based on the fluorometric determination of O_2_ and H^+^ levels with solid-state probes on a sensor cartridge. The oxygen consumption rate (OCR) was measured successively in the presence of RC substrates and then after the addition of ADP (phosphorylating respiration), ATP synthase F^o^-subunit inhibitor oligomycin (proton leak), oxidative phosphorylation uncoupler FCCP (maximal respiration), and the corresponding inhibitors of respiration chain complexes (non-mitochondrial respiration) (Figure 5a).

Even nanomolar concentrations of compound **4i** produced a statistically significant increase in the phosphorylating respiration and maximal respiration (Figure 5b,c) in the presence of RC complex I substrates (glutamate and malate) and in the presence of complex I inhibitor (rotenone) (Figure 5d). This finding corresponds to the above-described effect of the repolarization of mitochondria and also reflects the effect of bypassing the rotenone inhibition of ETC (Figure 4). A similar effect was previously observed for isolated brain mitochondria where **MB** increased the rate of cytochrome reduction using NADH as the electron donor [72,73].

The elucidated ability of compound **4i** to stimulate mitochondrial respiration and suppress the Ca^2+^-induced depolarization of mitochondria allows us to suggest that the new compound may possess cyto-(neuro)-protective properties and stimulate neuronal activity.

### 2.5. Evaluation of the Effect of Test Compounds on Binding of Specific Ligands of the Intrachannel and Peripheral Sites of the NMDA Receptor

The binding of all the new synthesized compounds as well as MB and MBH_2_ to glutamate receptors was studied by considering their ability to compete for binding with specific ligands of the NMDA receptor intrachannel and peripheral sites, which were radiolabeled [^3^H]MK-801 and [^3^H]ifenprodil, respectively. The experiments were carried out using a rat brain membrane fraction. Almost all the compounds showed no activity toward the NMDA receptor as the IC_50_ values were above 100 µM for both specific ligands. Methylene blue had a very low ability to interfere with MK-801 binding to the NMDA receptor (Table 5), but it effectively blocked the binding of [^3^H]ifenprodil. Only one compound of the whole series, **4i**, showed activity toward the NMDA receptor approaching that of **MB**.

### 2.6. Evaluation of the Effect of Lead Compounds **4i** and **4h** on the Assembly of Microtubules

Another important type of activity to evaluate in the test compounds is the stimulation of tubulin polymerization to give microtubules a normal structure. The compensation of the ability of tau protein to stabilize the microtubular structure, which has been lost as a result of pathological hyperphosphorylation and aggregation, may serve to normalize axonal transport and promote the growth of axons, thereby exerting therapeutic effects [74].

The **MB**-cycloalkaneindole conjugates were studied for their effect on tubulin polymerization. The anti-aggregation properties of **MB** toward tau protein have been ascribed to a relatively nonspecific oxidative mechanism involving the formation of covalent bonds between tau protein cysteine SH groups [75]. On the other hand, the tubulin dimer contains more than 20 cysteine residues, and a considerable fraction of them are significant for tubulin polymerization with an oxidation-dependent decrease of polymerization competence [76]. However, in our experiments, **MB** and **MBH_2_** induced a considerable concentration-dependent increase in the rate of polymerization, which was more than 1.5-fold at 100 µM (Figure 6). Therefore, the mechanism of potentiation of microtubule assembly by **MB** and **MBH_2_** might not be exclusively oxidative.

The same effect on tubulin polymerization seen with **MB** and **MBH_2_** was observed for lead compounds **4i** and **4h** (Table 6), although it was somewhat more pronounced.

It is important to note that straight, long microtubules with a normal structure were produced in the presence of **MB** compounds **4i** and **4h**.

### 2.7. Neuroprotective Effect of Lead Compound 4i under Calcium Overload Conditions on Primary Cultures of Rat Cerebellar Granule Cells

The above-described ability of the lead compounds to suppress the development of calcium-induced mitochondrial depolarization allows us to suggest that these compounds should have cyto(neuro)protective activity. For the validation of this suggestion, the influence of compound **4i** on calcium overload-induced death (in the presence of a calcium ionophore, ionomycin, where the cytoprotective effect may be connected only with intracellular targets, mainly with the mitochondria) was studied using a primary culture of rat cerebellar granule cells. This neurotoxicity model reflects the calcium stress involved with several types of insults, such as amyloid toxicity, excitotoxicity, and ischemic damage. The viability of the primary culture of rat cerebellar granule cells (CGC) and brain cortical neurons (BCN) was assessed by the MTT assay in the presence of various concentrations of compound **4i**. It was found that this compound did not affect formazan formation in the MTT assay. Thus, it could be inferred that it is not toxic to these cells and does not directly affect MTT conversion to formazan.

Compound **4i** produced a statistically significant protection from ionomycin-induced toxicity for CGC (Figure 7a) and BCN (Figure 7b).

## 3. Materials and Methods

### 3.1. In Vitro AChE, BChE, and CES Inhibition

Human erythrocyte AChE, equine serum BChE, porcine liver CES, acetylthiocholine iodide (ATCh), butyrylthiocholine iodide (BTCh), 5,5′-dithio-bis-(2-nitrobenzoic acid) (DTNB), 4-nitrophenol acetate (4-NPA), and BNPP were purchased from Sigma-Aldrich (St. Louis, MO, USA).

AChE and BChE activities were measured via the Ellman method [77]. The assay solution consisted of 0.1 M K/Na phosphate buffer pH 7.5, 25 °C with the addition of 0.33 mM DTNB, 0.02 unit/mL of AChE or BChE and 1 mM of substrate (ATCh or BTCh, respectively). The assays were carried out with a reagent blank containing all components (except AChE orBChE) in order to account for the non-enzymatic hydrolysis of the substrate. In addition, an enzyme blank was included that contained all components except substrate to account for non-substrate sulfhydryl groups.

The activity of CES was determined spectrophotometrically by monitoring the release of 4-nitrophenol at 405 nm [78]. The assay solution consisted of 0.1 M K/Na phosphate buffer pH 8.0, 25 °C, with the addition of 1 mM 4-nitrophenyl acetate and 0.02 unit/mL of CES. The assays were carried out with a blank containing all components, except CES.

The test compounds were dissolved in DMSO, and the incubation mixture contained 2% (*v*/*v*) of the solvent. Enzyme inhibition was first assessed at a single concentration of 20 µM for each compound after a 5 min incubation at 25 °C in three separate experiments. Compounds that inhibited the enzyme by more than 30% were then selected for the determination of the IC_50_ (the inhibitor concentration resulting in a 50% inhibition of control enzyme activity). Eight different concentrations of the test compounds in the range from 10^−11^ to 10^−4^ M were selected in order to obtain inhibition of AChE and BChE activity between 20% and 80%. The test compounds were added to the assay solution and preincubated at 25 °C with the enzymes for 5 min, followed by the addition of substrate. A parallel control was made for the assay solution with no inhibitor. Measurements were performed with a FLUOStar Optima microplate reader (BMG Labtech, Ortenberg, Germany). Each experiment was performed in triplicate. The results were expressed as the mean ± SEM. The reaction rates in the presence and absence of the inhibitor were compared, and the percent of residual enzyme activity due to the presence of test compounds was calculated. The IC_50_ values (the concentration of inhibitor required to decrease the enzyme activity by 50%) were determined graphically from the inhibition curves (the log inhibitor concentration vs. the percent residual enzyme activity) using Origin 6.1 software (OriginLab, Northampton, MA, USA).

### 3.2. Kinetic Analysis of AChE Inhibition. Determination of Steady-State Inhibition Constants

To elucidate the inhibition mechanisms for the active compounds, the residual activity of AChE was determined in the presence of three increasing concentrations of the test compounds and six decreasing concentrations of the substrates. The test compounds were preincubated with the enzyme at 25 °C for 5 min, followed by the addition of the substrates. Parallel controls were made for an assay of the rate of hydrolysis of the same concentrations of substrates in the solutions with no inhibitor. The kinetic parameters of substrate hydrolysis were determined. Measurements were performed with a FLUOStar Optima microplate reader. Each experiment was performed in triplicate. The results were fitted into Lineweaver–Burk double-reciprocal kinetic plots of 1/V versus 1/[S], and the inhibition constants *K_i_* (competitive component) and *αK_i_* (noncompetitive component) were calculated using Origin 6.1 software.

### 3.3. Propidium Iodide Displacement Studies

PIand donepezil were purchased from Sigma-Aldrich. The ability of the test compounds to competitively displacePI, which is a selective ligand of the peripheral anionic site of AChE, was evaluated with the fluorescent method [79,80]. As the source of the enzyme, electric eel (*Elecrophorus electricus*) AChE (*Ee*AChE, type VI-S, lyophilized powder, Sigma-Aldrich, Saint Louis, MO, USA) was used. The applicability of this enzyme was shown earlier [64]

To determine the percentage of displacement of PI from the *Ee*AChE PAS, *Ee*AChE (with a final concentration of 7 μM) was incubated with the test compound at a concentration of 3 and 20 μM in 1 mM Tris-HCl buffer pH 8.0, 25 °C for 15 min. Next, PIsolution (with a final concentration of 8 μM) was added, the samples were incubated for 15 min, and the fluorescence spectrum (530 nm (exc.) and 600 nm (emiss.)) was taken. Donepezil and decamethonium were used as reference compounds. The blank contained PI of the same concentration in 1 mM Tris-HCl buffer pH 8.0. The measurements were carried out in triplicate on a FLUOStar Optima microplate reader).

The percentage of displacement of PIfrom the peripheral anionic site of AChE was calculated using the following formula:% Displacement = 100 − (IF_AChE+Propidium+inhibitor_/IF_AChE+Propidium_) × 100,(1)
where IF_AChE+Propidium_ is the fluorescence intensity of the propidium associated with AChE in the absence of the test compound (taken as 100%) and IF_AChE+Propidium+inhibitor_ is the fluorescence intensity of the propidium associated with AChE in the presence of the test compound.

### 3.4. ABTS Radical Cation Scavenging Assay

The radical scavenging activity of the compounds was assessed using an ABTS radical decolorization assay. The procedure followed the reported protocol [65] with minor modifications [66]. ABTS (2,2ʹ-azino-bis-(3-ethylbenzothiazoline-6-sulfonic acid) diammonium salt) was purchased from TCI (Tokyo, Japan), while potassium persulfate (di-potassium peroxodisulfate), Trolox (6-hydroxy-2,5,7,8-tetramethylchroman-2-carboxylic acid), catechol, and DMSO were received from Sigma-Aldrich Chemical Co. (St. Louis, MO, USA). The ethanol was HPLC grade. Aqueous solutions were prepared using deionized water.

Trolox was used as the standard reference compound. Catechol was used as the positive control. All test and reference compounds were dissolved in DMSO. The final concentration of DMSO in the reaction mixture was 4% (*v*/*v*).

The solution of ABTS radical cation (ABTS^•+^) was produced by mixing a 7 mM ABTS stock solution with a 2.45 mM aqueous solution of potassium persulfate in equal quantities and allowing them to react for 12–16 h at room temperature in the dark. At the time of activity determination, the ABTS^•+^ solution was diluted with ethanol to adjust it to an absorbance value of about 0.80 ± 0.05 at 734 nm. A fresh working solution of ABTS^•+^ was prepared for each assay.

The radical scavenging capacity of the compounds was analyzed by mixing 10 μL of test compound with 240 μL of ABTS^•+^ working solution. The reduction in absorbance was measured spectrophotometrically at 734 nm using an xMark microplate UV/VIS spectrophotometer (Bio-Rad, Hercules, CA, USA). The reaction was monitored for an hour at 10 min intervals. Data were given for 1 h of incubation of compounds with ABTS^•+^. EtOH blanks were run in each assay. The results were obtained from three replicates of each sample and three independent experiments.

The antioxidant capacity as a Trolox equivalent (TEAC values) was determined as the ratio between the slopes obtained from the linear correlation of concentrations of test compounds and Trolox with absorbance of ABTS radical. For the test compounds, we also determined the IC_50_ values (the compound concentration required for a 50% reduction of ABTS radical). The compounds were tested in the concentration range of 1 × 10^−6^–1 × 10^−4^ M. The IC_50_ values were calculated using Origin 6.1 for Windows software (OriginLab, Northampton, MA, USA).

### 3.5. Oxygen Radical Absorbance Capacity Assay

The ORAC-FL method of Ou et al. [68], partially modified by Dávalos et al. [67], was followed, using a FluoStar Optima microplate reader with 485-P excitation and 520-P emission filters. 2,2´-Azobis(amidinopropane) dihydrochloride (AAPH), (±)-6-hydroxy-2,5,7,8-tetramethylchroman-2-carboxylic acid (Trolox), and fluorescein (FL) were purchased from Sigma-Aldrich. The reaction was carried out at 37 °C in 75 mM K,Na phosphate buffer (pH 7.4), and the final reaction mixture was 200 µL. The test compounds and the Trolox standard were dissolved in DMSO to 10 mM and further diluted in a 75 mM K,Na phosphate buffer (pH 7.4). The final concentrations were 0.1–1 µM for the test compounds and 1–6 µM for the Trolox. The blank was composed of 20 µL of a 75 mM K,Na phosphate buffer (pH 7.4) containing 20% (*v*/*v*) DMSO, 120 µL of FL and 60 µL of AAPH, and it was added in each assay. The antioxidant (20 µL) and FL (120 µL, final concentration: 70 nM) solutions were placed in a black 96-well microplate and were preincubated for 15 min at 37 °C. A solution of AAPH (60 µL, final concentration: 12 mM) was then added rapidly using a multichannel pipette. The microplate was immediately placed in the reader, and the fluorescence was recorded every minute for 100 min. The microplate was automatically shaken prior to each reading. The Trolox standard curve was also obtained in each assay. All reactions were carried out in triplicate, and at least three different assays were performed for each sample.

The antioxidant curves (fluorescence vs. time) were first normalized to the curve of the blank (without the antioxidant) corresponding to the same assay, and the area under the fluorescence decay curve (AUC) was calculated. The net AUC corresponding to a sample was calculated by subtracting the AUC corresponding to the blank. Regression equations were calculated by plotting the net AUC against the antioxidant concentration. The ORAC Tolox equivalent value (TE) was obtained by dividing the slope of the latter curve by the slope of the Trolox curve obtained in the same assay. The ability of Trolox to scavenge the peroxyl radical was equal to 1 [64,65]. Final ORAC values were expressed as μmol of test compounds per μmol of Trolox where the value of Trolox was taken as 1 [67,68]. The data are expressed as means ± SEM.

### 3.6. Lipid Peroxidation of Rat Brain Homogenate

On the day of the experiment, adult Wistar male rats that fasted overnight were euthanized in a CO_2_ chamber, followed by decapitation. The procedure was in compliance with the Guidelines for Animal Experiments at the Institute of Physiologically Active Compounds of the Russian Academy of Sciences. The brains were rapidly removed and homogenized in 0.12 M HEPES/0.15M NaCl, pH 7.4 buffer (HBS) (10 mg/g wet weight) and used immediately for the assay.

The protein concentrations in rat brain homogenate (RBH) were determined by the biuret assay using bovine serum albumin as the standard [81].

The effect of the compounds on LP of the RBH was studied at 30 °C for 40 min in 0.25 mL of RBH (2 mg of protein·mL^−1^) in the presence of the compounds or the vehicle (DMSO). Lipid peroxidation was induced by Fe^3+^ (0.5 mM Fe(NH_4_)(SO_4_)_2_) as an oxidant [82].

The reaction mixture was incubated for 30 min at 37 °C, then quenched by adding 0.4 mL of quench medium containing 250 mM HCl and 15% (*w*/*v*) trichloroacetic acid. The samples were then centrifuged for 10 min at 10,000× *g*, and 75 µL of the supernatant was transferred to a 96-well plate. Next, 75 µL of 0.8% (*w*/*v*) 2-thiobarbituric acid (TBA) was added. The plate was then sealed, heated at 95 °C for 15 min, cooled to 4 °C, and the absorbance at 530–620 nm was measured using a Wallac Victor 3 1420 Multilabel Counter (Perkin Elmer, Turku, Finland).

All the experiments were performed as four independent runs with different brain homogenate preparations and three repeated probes in each run. The average results for each run were normalized between the positive control with the vehicle and Fe^3+^ and negative control with only the vehicle. They were presented as the IC_50_ ± SD values for the antioxidant activities of the compounds which were calculated using GraphPad Prism 7.00 software (San Diego, CA, USA).

### 3.7. Rat Liver Mitochondria Isolation

Rat liver mitochondria were isolated by conventional differential centrifugation from the livers of adult Wistar strain rats that fasted overnight, pH 7.6 [83,84]. The mitochondrial protein concentration was determined using a biuret procedure with bovine serum albumin as the standard [81].

### 3.8. Mitochondrial Potential

Safranine O (10 µM) was used as a membrane potential probe [85]. Fluorescence intensity at 580 nm (excitation at 520 nm) was measured with a Wallac Victor 3 1420 Multilabel Counter (Perkin Elmer, Turku, Finland). The mitochondrial protein concentration was 0.2 mg/mL. The medium for measurements contained 75 mM sucrose, 225 mM mannitol, 10 mM K-HEPES (pH 7.4), 0.02 mM EGTA, and 1 mM KH_2_PO_4_. After a 5-min incubation, 5 mM glutamate/malate or 5 mM succinate in the presence of 0.5 µM rotenone were added to produce the mitochondrial potential. Then, the compounds (30 µM) or the same volume of the vehicle (DMSO) were injected into the mitochondrial suspension. After 15–20 min, 12.5 µM CaCl_2_ was added to each sample to induce the depolarization of mitochondria, and after 5 min, 0.5 µM CCCP was added for maximum depolarization of the mitochondria. The level of depolarization (ΔΨm) was calculated using the fluorescence value after 10-min incubation with 30 µM of compounds (or vehicle) normalized between the fluorescence measurements after substrate and CCCP addition, where fluorescence intensity at 580 nm was 100% after substrate addition and was 0% after CCCP addition.

### 3.9. Measurements of Oxygen Consumption in Rat Liver Mitochondria

The oxygen consumption was measured using a Seahorse XF96 flux analyzer on mitochondria respiring on glutamate (5 mM) and malate (5 mM), according the same protocol as [86] with a correction to a 96-well instead of a 24-well plate. Respiration by rat liver mitochondria (5 µg/well) was sequentially measured in a coupled state with substrates of complex I (basal respiration); followed by phosphorylating respiration after the addition of ADP; non-phosphorylating or resting respiration induced with the addition of oligomycin; uncoupled respiration stimulated by the addition of FCCP; and finally, non-mitochondrial respiration after the addition of the inhibitor of complex I rotenone.

### 3.10. Primary Screening of the Action of Compounds on Tubulin Polymerization

The assembly of tubulin into microtubules was carried out using pure tubulin from an HTS-Tubulin polymerization assay kit (Cytoskeleton, Inc., Denver, CO, USA). A standard polymerization reaction used 100 μL of 4 mg/mL tubulin in 80 mM PIPES (pH 6.9), 0.5 mM EGTA, 2 mM MgCl_2_, and 1 mM GTP. Polymerization was monitored by recording the change in absorbance on a Wallac Victor 3 1420 Multilabel Counter (Perkin Elmer, Turku, Finland) at λ = 3 55 nm. Electron microscopic monitoring was carried out with a Carl Zeiss Libra 120 electron microscope (Carl Zeiss Meditec AG, Jena, Germany) at 120 kV using negative contrasting.

### 3.11. Effect of Compounds on Ionomycin-Induced Toxicity in Primary Culture of Rat Cerebellar Granule Cells and Rat Brain Cortical Neurons

Cerebellar granule cells (CGC) and brain cortical neurons (BCN) were isolated from P-7 or P0-2 rat’s pups, respectively, using a previously described protocol [87]. Once removed, cerebella or brain cortex were digested for 15 min in 0.025% type II trypsin (*w*/*v*) at 37 °C. Ice-cold DMEM containing 5% (*w*/*v*) fetal bovine serum was added to inactivate the trypsin. The crude cell suspension was centrifuged, and the cell pellet was resuspended in Neurobasal with 2% (*v*/*v*) B27 supplemented with 15% (*v*/*v*) fetal bovine serum (FBS), 100 U/mL penicillin, and 100 mg/mL streptomycin. The cells were cultured in a humidified, 5% (*v*/*v*) CO_2_-controlled environment at 37 °C. After 8-10 day in culture the cells were incubated with a test compound or an equal volume of the DMSO (<1% (*v*/*v*) of the whole volume of the medium under the layer of cells and 3 µM ionomycin for 24 h. The cell viability was then evaluated as the dehydrogenase activity with the 3-(4,5-dimethylthiazol-2-yl)-2,5-diphenyltetrazolium bromide (MTT) assay. The absorbance was measured at 570 nm using a Victor microplate reader (Perkin Elmer), and data were normalized with intact control assumed 100%.

### 3.12. Radioligand Study of Interaction of the Compounds with NMDA-Receptor Binding Sites

The effect of test compounds on the radioligand binding to NMDA receptors was determined as previously described [4]. Two radioactive ligands were used: [^3^H] MK-801 (dizocilpine) with a specific activity of 210 Ci/mmol, binding to all isolated NMDA receptors, and [^3^H] ifenprodil with a specific activity of 79 Ci/mmol, binding only to NMDA receptors containing the NR2B subunit. A pellet of rat brain membrane specimen was resuspended in a working buffer (5 mM HEPES/4.5 mM Tris buffer, pH 7.6) in a ratio of 1:5 and stored in liquid nitrogen. The reaction mixture (with a final volume of 0.5 mL) contained 200 µL of the working buffer, 50 µL of 50 nM radioligand solution, and 250 µL of the membrane suspension. Nonspecific binding was determined in the presence of 50 µL of 1 M of unlabeled ligand.

For the binding study, the reaction mixture was incubated at room temperature for 2 h. After incubation, the samples were filtered through GF/B glass-fiber filters (Whatman), washed with the working buffer, dried, and transferred to scintillation vials, and then 5 mL of scintillation fluid was added to the vials containing 4 g of diphenyloxazole (PPO), 0.2 g of diphenyloxazoil benzene (POPOP), and 1 L of toluene. The radioactivity was determined in a TriCarb2800 TR scintillation counter (Perkin Elmer, Packard, Downers Grove, Illinois, USA) with a counting efficiency of about 65%.

The effect of the test compounds on the binding of [^3^H] MK-801 and [^3^H] ifenprodil to rat brain membranes was studied by adding 50 µL of the test compounds in the concentration range of 10^−8^–10^−3^ M to the incubation medium. From the results of inhibition, IC_50_ values were calculated for the test compounds using GraphPad Prism 4.

### 3.13. Molecular Modeling

#### 3.13.1. Structure Preparation

Geometries of the ligands were quantum-mechanically (QM) optimized in the Gamess-US package [88] using the B3LYP DFT method and the 6-31G * basis set (and 6-31++G ** for frontier orbital calculations). The partial atomic charges were taken from the QM results according to the Mulliken scheme [89]. These optimized geometries and partial charges were used for molecular docking.

There are several structures of human AChE *apo*-state and with several ligands available (PDB ID 4EY4-4EY8, [52]). All X-ray diffraction structures and a water-saturated optimized structure of *apo*-hAChE [51] were used for molecular docking (with water molecules removed). Similar to the results obtained in [51], the best binding affinities were obtained with structure PDB ID 4EY7 (hAChE co-crystallized with Donepezil, 2.35 Å), and they were similar to the results obtained with the water-saturated optimized structure of *apo*-hAChE.

#### 3.13.2. Molecular Docking

Molecular docking with a Lamarckian Genetic Algorithm (LGA) [90] was performed with Autodock 4.2.6 [91]. The grid box for docking included the whole active site and the gorge of AChE (22.5 Å × 22.5 Å × 22.5 Å grid box dimensions) with a grid spacing of 0.375 Å. The main LGA parameters were 256 runs, 25 × 10^6^ evaluations, 27 × 10^4^ generations, and a population size of 300. Figures were prepared with PyMOL (Schrödinger, New York, NY, USA).

## 4. Conclusions

The studied compounds effectively inhibited AChE and were shown to be potential blockers of AChE-induced β-amyloid aggregation. The high radical-scavenging activity of **MB**-cycloalcaneindole conjugates displayed in the ABTS and ORAC tests may be one of the main reasons for their high inhibitory activity of Fe^3+^-induced LP in rat brain homogenate.

Compound **4i** (R=CF_3_O, R_1_=CH_3_) showed properties similar to those of **MB** toward the ifenprodil-specific site of the NMDA receptor, ligands of which exert cognition-enhancing effects.

As a result of assessing the effect of conjugates **4** on the functions of mitochondria, two lead compounds, **4i** (R=CF_3_O, R_1_=CH_3_) and **4h** (R=CF_3_O, R_1_=H), were identified.

Mitochondrial functions studies showed that like **MB**, lead compound **4i** was able to decrease the efficiency of the inhibition of respiratory chain complex I, and, even in nanomolar concentrations, it stimulated the respiratory chain activity for isolated rat liver mitochondria. In addition, it could prevent calcium-induced mitochondrial depolarization, which suggests inhibition of the mitochondrial permeability transition and, accordingly, the possible presence of cyto(neuro)protective properties. The neuroprotective effect was confirmed using a cellular calcium overload model of neurodegeneration. An equally important feature of compounds **4i** and **4h** was their ability to stimulate the assembly of microtubules. This finding suggests a compensatory effect on the system of microtubules in the case of tauopathy, which is one of the most pronounced pathogenic characteristics of neurodegenerative diseases.

Thus, new **MB**-cycloalkaneindoles conjugates form a promising class of compounds for the development of multitarget neuroprotective drugs, which simultaneously act on several targets, thereby providing cognitive stimulating, neuroprotective, and disease-modifying effects.

## Data Availability

Not applicable.

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
