# Peer review of "Conjugates of Methylene Blue with Cycloalkaneindoles as New Multifunctional Agents for Potential Treatment of Neurodegenerative Disease"

_ijms, 2022, doi:10.3390/ijms232213925_

Round 1

Reviewer 1 Report

The manuscript ijms-1977766 "Conjugates of methylene blue with cycloalkaneindoles as new multifunctional agents for potential treatment of neurodegenerative disease." by Bachurin et al. describes the study of biological activity of previously synthesized conjugates of methylene blue (MB) with cycloalkaneindoles. The results obtained by the authors show that these compounds can be used to develop multipurpose neuroprotective drugs which act simultaneously on several targets. The authors have interesting results, so I think that this paper will be of interest to the readers of IJMS.

Questions and comments:

1) Why were three compounds (4m, 4c, and 4i) chosen for molecular docking, but the mechanism of AChE inhibition was established for only two of them?

2) Part 2.5. These data were partially published earlier [ref. 33], however, the data on the binding of radioligands [3H]MK-801 differ from previously published results. How can the authors comment on this?

3) What about the toxicity of the obtained compounds?

4) I recommend comparing the results obtained by the authors with previous results obtained by other scientific groups.

5) I recommend the authors to strengthen the Introduction part about synthesis and applications of MB derivatives. Recent review articles of 2022 on this topic should be added. For example, 10.1016/j.dyepig.2022.110806, 10.3390/molecules27010276, 10.1134/S0006297922090073.

6) Minor changes:

- line 75. Please delete the underline "has been".

- Table 1. The presence of absolute and percentage values in the same table looks confusing. Please also check rounding to significant figures.

- I propose to combine all the sentences in the conclusions into one paragraph.

- Please recheck Russian symbols such as "с, м, в, н, а, е, т" in the text of the manuscript.

Reviewer 2 Report

The manuscript “Conjugates of methylene blue with cycloalkaneindoles as new 2 multifunctional agents for potential treatment of neurodegenerative disease” from Bachurinet al., are well written, with good English grammar. Authors report that a new MB-cycloalkaneindole conjugates constitute a promising class of compounds for the development of multitarget neuroprotective drugs, which simultaneously act on several targets, thereby providing cognitive stimulating, neuroprotective, and disease-modifying effects. This is a really interested study, but major and minor revision will be necessary.

Major revision

1. Can you explain over your results why you think one compound of the whole series, 4i, showed activity toward the NMDA receptor

2. Could you explain why you think that only the compound 4i showed activity toward the NMDA receptor and the others don’t? You suggest that may possess cyto-(neuro)-protective properties and stimulate neuronal activity, but you are using isolated rat liver mitochondria and will be better if you use isolated rat brain mitochondria. It will be logic. This referee suggests the use of brain mitochondria.

3. Authors study the viability, using MTT assay, of primary culture of rat cerebellar granule cells. As authors know, cerebellar cells are not involved in AD. It is the cells of the cortex or hippocampus, among other areas of the brain, that are involved. It is also in these areas where amyloid accumulations appear, and neurons die due to the presence of TAU phosphorylation. The authors should have used primary cultures of cerebral cortex or hippocampal cells. This referee considers it to be a priority.

4. Authors indicated that the mechanism of potentiation of microtubule assembly by MB and МВН2 might not be exclusively oxidative. Can you suggest other mechanisms, like inflammation by amyloid proteins or something?

Minor revision:

Introduction 1. Take out spaces between words when them appear.

2. Put one space when it appears.

Round 2

Reviewer 1 Report

I thank the authors for answering my questions and improving the manuscript.

Reviewer 2 Report

Accept for publication